# One-Dimensional Phononic Crystals: A Simplified Platform for Effective Detection of Heavy Metals in Water with High Sensitivity

**DOI:** 10.3390/mi14010204

**Published:** 2023-01-13

**Authors:** Abdulkarem H. M. Almawgani, Hamza Makhlouf Fathy, Ghassan Ahmed Ali, Hussein A. Elsayed, Ahmed Mehaney

**Affiliations:** 1Electrical Engineering Department, College of Engineering, Najran University, Najran 61441, Saudi Arabia; 2TH-PPM Group, Physics Department, Faculty of Science, Beni-Suef University, Beni-Suef 62512, Egypt; 3Information Systems Department, College of Computer Sciences and Information Systems, Najran University, Najran 61441, Saudi Arabia

**Keywords:** heavy metals, cadmium bromide (CdBr_2_), phononic crystals, sensor, acoustic waves, sensitivity, band gab

## Abstract

Recently, the pollution of fresh water with heavy metals due to technological and industrial breakthroughs has reached record levels. Therefore, monitoring these metals in fresh water has become essentially urgent. Meanwhile, the conventional periodic one-dimensional phononic crystals can provide a novel platform for detecting the pollution of heavy metals in fresh water with high sensitivity. A simplified design of a defective, one-dimensional phononic crystals (1D-PnC) structure is introduced in this paper. The sensor is designed from a lead-epoxy multilayer with a central defect layer filled with an aqueous solution from cadmium bromide (CdBr_2_). The formation of a resonant peak through the transmittance spectrum is highly expected. This study primarily aims to monitor and detect the concentration of cadmium bromide in pure water based on shifting the position of this resonant peak. Notably, any change in cadmium bromide concentration can affect the acoustic properties of cadmium bromide directly. The transfer matrix method has been used to calculate the transmission spectra of the incident acoustic wave. The numerical findings are mainly based on the optimization of the cadmium bromide layer thickness, lead layer thickness, epoxy layer thickness, and the number of periods to investigate the most optimum sensor performance. The introduced sensor in this study has provided a remarkably high sensitivity (S = 1904.25 Hz) within a concentration range of (0–10,000 ppm). The proposed sensor provides a quality factor (QF), a resolution, and a figure of merit of 1398.51752, 48,875,750 Hz, and 4.12088 × 10^−5^ (/ppm), respectively. Accordingly, this sensor can be a potentially robust base for a promising platform to detect small concentrations of heavy metal ions in fresh water.

## 1. Introduction

Phononic crystals (PnCs) are artificial periodic structures, which exhibit sound wave propagation. The phononic bandgap presents the most notable property of PnCs due to its mastery of the confinement of the sound wave from propagating through these PnC structures [1,2]. The mechanical properties of the PnCs constituent materials represent a key motivator to the tenability of these gaps. Due to their abilities to regulate not just sound waves but also the complete spectrum of mechanical waves from the audible range (1–20 kHz) to THz frequencies, these artificial periodic structures have recently received a lot of attention [3,4,5]. Because PnCs can influence the propagation of acoustic waves, their effective use and application in a wide range of engineering applications have become noticeably popular. In addition, the breaking of PnCs periodicity by the inclusion of some disorders or defect layers expands the range of applications based on these structures. In this context, PnCs are utilized in many devices and applications, including acoustic cloaking, sensors, temperature sensors, filter multiplexing systems, liquid sensors, Waveguides seismic wave reflection, actuators, heat isolation systems, and the development of acoustic Metamaterials [6,7]. The development of acoustic sensors as a novel framework for liquid sensing purposes is one of the most basic applications of PnCs. Sensors constructed on PnCs can be used to detect and sense the type, concentration, and physical properties of liquids [8,9,10]. In this regard, Zubtsov et al. constructed a PnC sensor that determines the concentration of a water/1-propanol solution [11]. Moreover, a sensitive biosensor using a two-dimensional triangular lattice solid/fluid phononic crystal (PnC) to detect the temperature of Methyl Nonafluorobutyl Ether (MNE) within the range of 10–40 °C is presented [12]. Lucklum et al. have also developed a new sensor platform based on a PnC cavity for determining the characteristics of gasoline [13].

Nowadays, some studies have focused on the use of PnCs in sensing and detecting liquid type, concentration, and characteristics from various solutions. For example, the design of PnCs as a demultiplexer can be used as a selective sensor to separate three harmful heavy metals (CuSO_4_, MgSO_4_, and MnSO_4_) from water at a very low concentration of 1% [14]. Therefore, PnC can detect and quantify heavy metals very easily. This study mainly aims to present a simple design based on a defective 1D-PnC as a sensor, especially for heavy metals such as Cadmium bromide. Heavy metals can have a density five times greater than that of fresh water. These elements can have a variety of harmful effects on humans, and their toxicity is proportional to their mass. As a result, environmental pollution from heavy metals has recently given rise to more ecological and global health issues [15,16,17]. Even though these metals exist naturally and in small amounts throughout the earth’s crust, human exposure is primarily caused by anthropogenic activities, such as smelting, mining, agricultural, and industrial operations [18,19,20]. Heavy metals can enter the body by inhalation and ingestion once they have been released into the environment. Additionally, metals accumulate in body tissues at a rate that is greater than what the body can handle throughout the detoxification process, leading to a gradual buildup [21,22,23].

Cadmium bromide is very harmful to people’s health, and it might be exposed to humans and animals at work or in the environment as a by-product of zinc manufacturing. Humans are commonly exposed to this metal by inhalation and ingestion and can develop acute and chronic poisoning. Cadmium bromide is an inorganic pollutant and a synthetic compound, which is used in photography, lithography, and engraving [22,23,24,25]. Cadmium ranks as the seventh most dangerous heavy metal, according to the ATSDR agency [24,25,26]. Cadmium is a zinc analog with a strong affinity for it, and it can interfere with its biological functions. It also binds and activates the estrogen receptor, generating estrogenic effects, such as reproductive dysfunction and perhaps encouraging the growth of some types of cancer cells [27,28,29,30]. Cadmium activates mitogen-activated protein kinases, which leads to cell death. Bromine is a strong oxidizing agent that can liberate oxygen-free radicals from mucous membrane water. These free radicals are also powerful oxidizers that harm tissues. Moreover, an additional irritation will arise from the creation of hydrobromic and bromic acids. Bromism is a condition caused by the bromide ion, which affects the central nervous system [28,29,30]. Based on the discussion above, cadmium bromide is very toxic and should be banned. Therefore, it is very useful if the concentration of cadmium bromide is measured by using a new, cost-effective, high-sensitivity sensor that is capable of measuring even the smallest variations in its concentration.

In this study, the effects of different concentrations of an aqueous solution of cadmium bromide on the transmission spectra of a 1D PnC sensor are investigated. The transfer matrix method (TMM) is used for calculating the transmission coefficient of the incident acoustic waves. To distinguish between the cadmium bromide concentrations in parts per million (ppm), a binary multilayer defective PnC structure has been devised. For each concentration, the sensor performance characteristics in the vicinity of sensitivity (S), the quality factor (Q), and the figure of merit (FOM) were all calculated and tabulated accordingly.

## 2. Theoretical Analysis and Design

### 2.1. Model Design

The suggested 1D binary PnC sensor structure is shown in Figure 1. The structure is configured from an infinite number of unit cells that are intermediated by a defect layer ($d_{f}$) filled with cadmium bromide. Therefore, the proposed design is labeled as ([lead/epoxy]^N^(defect layer) [lead/epoxy]^N^), whereby N defines the number of unit cells, in which each unit cell is made up of two materials, lead, and epoxy, with thicknesses of $d_{1}$ and $d_{2}$, respectively. In addition, a = $d_{1}$ + $d_{2}$ is the thickness of each unit cell. The selection of these materials is based on their wide-ranging uses in the field of phononic crystals [3,5]. The acoustic properties of the utilized materials are listed in Table 1 [3,5].

### 2.2. Theoretical Treatment

Over the past three decades, TMM is considered the best theoretical tool for describing acoustic wave interactions with PnC structures. A brief description of this method is presented in this paper to demonstrate the transmission spectrum of the suggested structure regarding the interaction with incident acoustic waves. First, as illustrated in Figure 1, a single unit cell of a 1D PnC is exposed to the incident acoustic waves. Then, a generalized formula can be released for the whole structure. This interaction is restricted to the *x*-axis only, and each unit cell has two layers of lead and epoxy with thicknesses of d_1_ and d_2_, respectively. The governing equation of the normal incident of the acoustic wave on the PnC structure is presented as follows:(1)$$\nabla^{2}\gamma =C_{j}^{2}\ddot{\gamma }$$
where γ is the displacement potential and $C_{i}=\sqrt{\frac{\lambda +2\mu }{\rho }}$ is the acoustic wave velocity in each layer [lead, epoxy], λ,μ are Lame’ coefficients, and the subscript *j* = 1, 2. Then, Equation (1) can be solved as follows:(2)$$\gamma =Xe^{i\left(\omega t-k_{j}x\right)}+Ye^{i\left(\omega t+k_{j}x\right)}$$
where $i^{2}=-1,k_{j}=\omega \sqrt{\frac{\rho_{j}}{C_{xxx}^{j}}}$ is the wavenumber in each layer,$\;\rho_{j}$ is the mass density, ω is the angular frequency,$\;C_{xxx}^{j}$ describes the elastic stiffness constant of a distinct layer *j*, and *X* and *Y* are two arbitrary coefficients [31,32,33]. The stress components and the dimensionless displacement of the incident acoustic wave can be written as follows:(3)$$\bar{\sigma_{x}}=\lambda \left(\frac{\partial^{2}\gamma }{\partial x^{2}}\right)+2\mu \left(\frac{\partial^{2}\gamma }{\partial x^{2}}\right)$$
(4)$$\bar{v_{x}}=\frac{\partial \gamma }{\partial x}$$

Therefore, at the right and left sides of the layer *j* in the *k* the unit cell, the two-state vectors, which represent the full acoustic wave propagation, are given as follows:(5)$$V_{jL}^{\left(k\right)}=\left\{{\bar{\sigma }}_{xjL}^{\left(k\right)},{\bar{v}}_{xjL}^{\left(k\right)}\right\}$$
(6)$$V_{jR}^{\left(k\right)}=\left\{{\bar{\sigma }}_{xjR}^{\left(k\right)},{\bar{v}}_{xjR}^{\left(k\right)}\right\}$$
where the subscripts *R* and *L* refer to the right and left sides of layer *j*. The relationship between the right and left state vectors of the layer *j* in the *k*th unit cell can be presented as follows:(7)$$V_{jR}^{\left(k\right)}=T_{j}^{\prime }V_{\mathrm{jL}}^{\left(k\right)}$$
where T*_j_*’ is a 2 × 2 transfer matrix, and its elements are described as follows:(8)$$T_{j}^{\prime }\left(1,1\right)=T_{j}^{\prime }\left(2,2\right)=\frac{\left[\exp\left(-iq_{Lj}x_{j}\right)+\exp\left(iq_{Lj}x_{j}\right)\right]}{2}$$
(9)$$T_{j}^{\prime }\left(1,2\right)=\frac{iq_{Lj}\left(\lambda +2\mu \right)\cdot \left[\exp\left(iq_{Lj}x_{j}\right)-\exp\left(-iq_{Lj}x_{j}\right)\right]}{2}$$
(10)$$T_{j}^{\prime }\left(2,1\right)=\frac{i\left[\exp\left(iq_{Lj}x_{j}\right)-\exp\left(-iq_{Lj}x_{j}\right)\right]}{2q_{Lj}\left(\lambda +2\mu \right)}$$

Hence, the relationship between two consecutive state vectors in the *k*th and (*k*−1) unit cells is defined as [34,35,36]:(11)$$V_{2R}^{\left(k\right)}=T_{k}V_{2R}^{\left(k-1\right)}$$
where T*_k_* is a transfer matrix that combines two successive unit cells and can be described in the following form:(12)$$T_{k}=T_{2}T_{1}^{\prime }$$

Therefore, the transmission coefficient of the incident acoustic wave via the PnC can be presented as follows:(13)$$\frac{U_{e}}{U_{0}}=\frac{2E_{0}\left(T_{11}T_{22}-T_{12}T_{21}\right)}{E_{0}\left(T_{11}-E_{e}T_{21}\right)-\left(T_{12}-E_{e}T_{22}\right)}$$
where $E_{0}$ and $E_{e}\;$are Young’s moduli of the two semi-infinite solids at the left and right of the structure.${\;U}_{0}$ and $U_{e}$ define the amplitudes of the transmitted and incident acoustic wave, and $T_{\mathrm{ij}}\;$represents the elements of the total transfer matrix$\;T=T_{n}T_{n-1}\ldots T_{k}\ldots T_{1}$.

## 3. Results and Discussion

### 3.1. The Acoustic Properties of Cadmium Bromide

Firstly, the role of changing the concentrations of CdBr_2_ aqueous solution versus its acoustic properties (density and sound velocity) has been described based on the demonstrated experimental data provided by [35]. Then, the numerical fitting for this data has been examined to generalize the relationship between concentrations and acoustic properties. The experimental data for the mass density of CdBr_2_ are fitted to the following fitting equation:(14)$$\rho \left(\frac{\mathrm{kg}}{m^{3}}\right)=\alpha +b\times C\;\left(\mathrm{ppm}\right)$$
where ρ refers to the density of CdBr_2_, C refers to its concentration, and α and b are the coefficients of the fitted relationship, whereby α = 1.00104 and b = 8.09362 × 10^−7^. As shown in Figure 2, an increase has been observed in the density of CdBr_2_ with increasing its concentration, which is a linear fitting according to the previous equation. The concentration of a solution is related to its density. In other words, the density of a solution is proportional to its concentration. Therefore, when the concentration of a solution increases, the density of the solution also increases. When the concentration of a solution rises, the number of solute molecules that dissolve in the solvent rises as well. Thus, it can be concluded that the solution has a higher mass per unit volume.

The experimental data for the sound speed of CdBr_2_ are fitted in the following equation:(15)$$v\left(\frac{m}{s}\right)=\beta_{1}-\beta_{2}\times C+\beta_{3}\times C^{2}$$
where v describes the sound speed of CdBr_2_, C defines its concentration, and $\beta_{1},\beta_{2}\;,{\mathrm{and}\;\beta }_{3}$ are the coefficients of the fitted relationship. The values of these coefficients are given as:${\;\beta }_{1}=1506.99251,\;\beta_{2}=2.75034\times 10^{-4}\;{\;\mathrm{and}\;\beta }_{3}=1.80484\times 10^{-4}$. Figure 2 demonstrates that the sound velocity decreases with the increase of the CdBr_2_ concentration, leading to a polynomial linear fitting according to the previous equation. A variable-path interferometer is an experimental device, which is utilized for investigating the ultrasonic velocity as a function of concentration in aqueous solutions of cadmium halides [34,35]. For cadmium chloride, the ultrasonic velocity increases linearly with concentration, unlike the cases of cadmium bromide and cadmium iodide. The hefty bulk of the ions in the solution has been attributed to this feature [34,35]. For cadmium chloride, the apparent molar compressibility changes linearly with the square root of concentration. The linear relationship does not hold for cadmium bromide and iodide. The measurements of activity coefficients, transport numbers, and other thermodynamic parameters showed that cadmium halides deviated significantly from the normal behavior of electrolytic solutions. They are primarily found in solutions as auto complexes, such as Cd[CdI_3_]_2_. In the presence of alkali halides, they also form stable complex ions [34,35].

### 3.2. Transmission Spectrum of the Defective Structure of Cadmium Bromide

In this subsection, the numerical results in the vicinity of the transmission spectrum of the PnC structure are presented and described in Figure 1. Thus, the focus is on the optimization process of some parameters such as thickness and number of unit cells to test the highest sensitivity of the sensor, whereby the number of periods N = 2, the thickness of lead, epoxy and CdBr_2_ defect layers are $d_{1}=d_{2}=d_{f}=$ 0.5 × 10^−6^ m. The PnC sensor design will be as ([lead/epoxy]^2^(CdBr_2_) [lead/epoxy]^2^). Figure 3 exhibits the transmittance of the structure in this study with and without the inclusion of a defect layer from CdBr_2_. Here, the transmission spectrum is plotted for the incident acoustic waves versus the normalized frequency (*ωa∕2λc*), where *c* is considered in all calculations as the speed of sound in epoxy.

Figure 3a simulates the transmission spectrum for the perfect design without a defect layer. On the other hand, Figure 3b describes the transmission spectrum of the symmetric design [(lead/epoxy)^2^ [CdBr_2_] (lead/epoxy)^2^]. Consequently, the concentration of CdBr_2_ through the defect layer is 0 ppm.

As shown in Figure 3a, a wide phononic bandgap without any transmitted peaks appeared in the normalized frequency, ranging between 5.1–5.3 (corresponding to the absolute frequency in the range of 6.47445 × 10^10^—6.72835 × 10^10^ Hz). Because there is no defect layer, the bandgap is large and flat, with no transmitted peaks. The constructive interference of the acoustic waves at the interfaces of the constituent materials caused the formation of this bandgap. By including a defect layer filled with cadmium bromide, the bandgap’s left and right edges were slightly affected, as shown in Figure 3b. Moreover, a sharp resonant or transmitted peak was formed at a normalized frequency of 5.1885 (f= 6.5868 × 10^10^ Hz). The amount of acoustic energy transmitted across the bandgap refers to the transmitted peak. It is an important characteristic property of PnCs, which occurs when the defect layer is filled with fluids, as the resonance frequency is connected to the fluid material’s acoustic properties. Therefore, with different cadmium bromide concentrations, the resonance frequency value could be shifted. To obtain a high sensitivity of the proposed sensor, an optimization for each parameter was carried out to select the optimum parameters for the designed sensor.

### 3.3. Optimization of Sensor Design

The optimization for the input parameters such as the thickness of lead, epoxy, cadmium bromide, and the number of unit cells ($d_{1},\;d_{2},d_{f}\;\mathrm{and}\;N)$ are described in this sub-section. As mentioned above, the purpose of this scenario is to obtain the optimum values of these parameters to achieve the most efficient performance for the proposed sensor.

#### 3.3.1. Optimization of the Cadmium Bromide Layer Thickness

Figure 4 shows the effect of the thicknesses of the cadmium bromide layer on the sensitivity of the proposed design due to varying concentrations, ranging from 0 to 10,000 ppm. The number of periods, the thickness of the lead layer, and the thickness of the epoxy layer were considered as listed previously. Figure 4 displays a remarkable effect of the thickness of the analyte layer on the obtained sensitivity values. The highest sensitivity (375 Hz/ppm) is obtained at $d_{f}=$ 0.1 × 10^−6^ m. At $d_{f}=$ 0.2 × 10^−6^ m, the structure sensitivity is close to zero. The unusual decrease in the sensitivity value at a thickness of 0.2 μm can be attributed to the condition of constant phase shift at this particular thickness value. This explanation is physically based on Bragg’s diffraction law, which is considered the main core of the band gap and resonant modes formation [3,36]. At this definite value, sensitivity dropped to approximately zero, underlying that the resonance peak position was almost unchanged at this thickness value. Based on the above-mentioned Bragg’s diffraction law, at this thickness value, the same type and order of interference occurred at this value and, therefore, the peak position did not experience any change.

For thicknesses larger than 0.2 × 10^−6^ m, sensitivity was almost unaffected. Such a response may provide some flexibility in considering the thickness of the defect layer from a manufacturing point of view. However, the large values of the defect layer thickness are accompanied by the appearance of two or more resonant peaks, which could lead to some disruptions during the detection process. In addition, one resonant peak is preferred in the field of sensors as a general case [5,9,10]. Therefore, a value of 0.1 μm is considered as the optimum thickness of the defect layer.

#### 3.3.2. Optimization of the Lead Layer Thickness

Figure 5 plots the effect of the thickness of lead layers on the sensitivity of the proposed design as the concentration of cadmium bromide varied from 0 to 10,000 ppm. Figure 5 shows a significant decrease in the values of sensitivity from 625 to 250 Hz/ppm as the thickness of lead layers increased from 0.1 × 10^−6^ m to 1 × 10^−6^ m, respectively. This response can be attributed to the lead layer’s considerable thickness, thereby increasing the losses or attenuation of the incident waves. Therefore, the interaction of these waves with the 1D-PnC and the formation of resonant peaks are very weak. Hence, the small value of lead layer thickness is essentially required to stimulate the resonant peaks. Therefore, selecting a small value of the lead layer greatly enhanced the performance of the designed sensor.

#### 3.3.3. Optimization of the Epoxy Layer Thickness

The impact of the epoxy layer thickness on the sensitivity of the proposed design is shown in Figure 6. There is a considerable loss in sensitivity as the thickness of the epoxy layers increased from 0.1 × 10^−6^ to 1 × 10^−6^ m. For d_1_ = 0.1 × 10^−6^ m, sensitivity recorded the value of 1905 Hz/ppm, and at d_1_ = 1 × 10^−6^ m, it dropped to 350 Hz/ppm. The reason for such sensitivity decrement is the same reason for the effect of lead layer thickness. The only difference between them is that the value of the sensitivity decrement in the case of epoxy is smaller than lead, and this difference is certainly due to the small acoustic impedance of epoxy than lead ($z=\sqrt{\rho v}$) [1,4].

#### 3.3.4. Optimization of the Number of Unit Cells

Finally, the optimization procedure has been accomplished by studying the role of the number of unit cells in the proposed sensor’s sensitivity. The influence of the number of periods of the 1D-PnCs on the sensitivity of the new design is shown in Figure 7. The thicknesses of the constituent materials are $d_{1}$ = $d_{2}$ = 0.1 × 10^−6^ m and $d_{f}$ = 0.1 × 10^−6^ m, respectively. As the defect layer is sandwiched between two identical unit cells, the position of the resonant peak is almost unchanged with varying cadmium bromide concentrations. Thus, the structure sensitivity is close to zero. As the number of periods (N) increased to 2, the sensitivity achieved a maximum value of 1904 Hz/ppm, as shown in Figure 7. The increase in the number of periods enhanced the interferences of acoustic waves at interfaces, as well as the displacement of the formed resonant peaks [4]. However, the displacement of these modes suffered from increasing the total impedance of the crystals. Therefore, sensitivity reached saturation; then, it started to decrease as shown in Figure 7. Based on the results obtained in Figure 4, Figure 5, Figure 6 and Figure 7, it is believed that the best condition for the sensor design is a symmetric binary structure with several unit cells N = 2 as [(lead/epoxy)^2^ [cadmium bromide] (lead/epoxy)^2^], defect layer thickness = 0.1 × 10^−6^ m and $d_{1}$ = $d_{2}\;$ = 0.5 × 10^−6^ m.

### 3.4. The Effect of Cadmium Bromide Concentration

This subsection discusses the effects of increasing the concentrations of cadmium bromide on the characteristics of the resonant peak at optimum values. In Figure 2, different concentrations of the sound speed and density of cadmium bromide solution are shown and the effect of these concentrations (10, 40, 60, and 90 × 10^3^ ppm) on the PnC structure is considered. By considering different concentrations of cadmium bromide ions in pure water, the resonance peak in Figure 3 has shifted to new positions, as shown in Figure 8. Based on this figure, the frequency of the resonant peaks changed from 658,680 to 658,362 × 10^5^ Hz with changing cadmium bromide concentrations within the range of 0 and 90,000 ppm. This is because any increase in the concentration of cadmium bromide leads to an increase in the density of cadmium bromide and a decrease in its sound speed as demonstrated in Figure 2. Therefore, the position of the resonant peaks changed as well. It has been observed that the sensitivity increased from 846.33 to 1904.25 Hz with a constantly gradual increase of the cadmium bromide concentration between 0 and 90,000 ppm, respectively. Figure 9 shows a color map that simulates the appearance of the resonant peak within the formed phononic band gap due to an increase in the concentration of cadmium bromide and the movement of the resonant peak through the phononic band gap. To recap, the transmission distribution of the acoustic waves in the defected PnC versus the normalized frequency for a concentration range of 0–90 × 10^3^ ppm is shown in Figure 9. The illuminated spots (the highest transmission values; resonant modes) within the normalized frequency range of (5.18–5.2) shifted with increasing different concentrations. This means that the illuminated line has a very clear slope and the normalized frequency values (indicated on the *x*-axis) changed with the use of varying concentrations (as indicated on the left *y*-axis), which, in turn, confirmed the obtained results in Figure 8. In addition, different shades of colors indicated different percentages of the intensity of the transmitted waves and the high sensitivity of the proposed sensor versus the concentration in the ppm scale. Furthermore, the detailed results of cadmium bromide sensitivity and resonance frequency for all the utilized concentrations are listed in Table 1 under the supplementary data.

### 3.5. Analysis of the Proposed Sensor’s Performance

This section discusses the performance of the proposed sensing tool based on certain associated criteria, including sensitivity (S), quality factor (QF), and figure of merit (FOM). These factors are thought to be of potential significance in describing the sensor’s performance. Their values are mostly based on defect mode characteristics. The sensor’s sensitivity is defined as the change in the position of the defect mode as a function of concentration variation. Thus, the following relationship can be used to determine its values [37,38]:(16)$$S=\frac{\Delta f}{\Delta C}$$
where Δf denotes the resonance peak or transmitted frequency for each concentration, and ΔC denotes the change in the cadmium bromide concentration. In addition, other related performance parameters were calculated, such as quality factor (QF), which is greatly influenced by the position of the resonant peak [37,38] as follows:(17)$$Q=\frac{f_{r}}{f_{HBW}}$$
where $f_{r}$ is the resonant peak frequency, and $f_{HBW}$ is the half bandwidth frequency of this peak [14]. A high-quality factor indicates sharp resonant peaks, which increases the frequency resolution. A parameter was also obtained; it is known as the figure of merit (FOM), which predicts the sensor’s ability and efficiency in measuring any change in the resonant frequency. The formula for calculating FOM is as follows [39,40,41]:(18)$$\mathrm{FOM}=\frac{s}{F_{HBW}}$$

FOM increases as the half bandwidth frequency decreases. Finally, another performance parameter (i.e., the damping rate) was also considered. The damping, which defines how the acoustic waves in the designed PnCs decay following a disturbance across the structure, determines the sharpness of the resonance-transmitted peaks [42,43,44,45]:(19)$$\zeta =\frac{1}{2*Q}$$

Figure 10 exhibits the effect of different concentrations on the resonance peaks of the cadmium bromide liquid sensor, which, in turn, affects sensitivity. With increasing the concentration of cadmium bromide, the resonant modes are linearly shifted toward higher frequencies, and the sensor’s sensitivity increases as well. Therefore, it can be guaranteed that the proposed PnC sensor provided high sensitivity and performance for the concentration range from zero to 90,000 ppm. Figure 10 indicates that the suggested design provides a relatively high sensitivity of 1904.25 Hz/ppm as the concentration of cadmium bromide increases from 0 to 10,000 ppm. This value gradually decreased with the increase of cadmium bromide concentration until it reached 846.333Hz/ppm at 90,000 ppm. The investigated sensitivity of the proposed design could be promising compared to the results of previous studies [40,41]. The sensors, which were previously introduced, are based on 1D PnC designs, which provide a maximum value of 969,973 Hz due to the concentration variations in mol/L.

Figure 11 introduces the effect of the damping rate of the acoustic waves and the quality factor of the proposed sensor in this study. Figure 11 shows that the maximum value of QF is demonstrated at the minimum value of the damping rate. Equation (19) proves an inverse proportion between QF and ζ. The high values of QF are obtained due to the lower values of FWHM. Therefore, small values of the damping rate are investigated due to these small values of the FWHM. On the other hand, small FWHM values indicated a limited possibility of the designed structure absorbing the incident acoustic waves. As the damping rate is low, the sharpness of the resonance peak increases. As observed in Figure 11, when the concentration is equal to the value of 90,000 ppm, this is the highest sharpness of the resonance peak represented by the Q factor of the value of 1528.76 since it has the lowest damping rate of the value (0.33 × 10^−7^). However, when the concentration of CdBr_2_ is equal to the value of 30,000 ppm, the peak has the lowest sharpness, (Q) of the value of 1339.92, and it has the highest damping rate of the value of 0.37 × 10^−7^. Nevertheless, in the entire CdBr_2_ concentrations, the quality factor is very high as it changed from 1339.92 to 1528.76. This indicates that all the resonant peaks are sharp, which increased the frequency resolution of the proposed sensor. In comparison to other sensors, the numerical findings investigated in Figure 11 indicated some special features of the designed sensor compared to its counterparts in 1D PnC designs [42,43]. The quality factor of the proposed design sensor may rise to around 1528.76, which is relatively high compared to the results explored in previous 1D PnC sensors [42,43].

Figure 12 demonstrates the effect of different concentrations of CdBr_2_ on the detection limit and FOM of the cadmium bromide liquid sensor. As seen in this figure, when the concentration of cadmium bromide increases, the figure of merit decreases. It was found that the values of the figure of merit were shifted from 4.12088 × 10^−5^ to 1.96657 × 10^−5^ (/ppm) when the concentrations were changed from 10,000 to 90,000 ppm. The values of the figure of merit are small, but they are acceptable for small concentrations, which are measured about a part per million (ppm). In addition, FOM has a similar response, which is similar to the response obtained for the case of sensitivity as this parameter is defined as the reduced sensitivity. Figure 12 shows the relationship between different concentrations and the detection limit of the cadmium bromide liquid sensor. The limit of detection (LOD) has a lower concentration of an analyte in a sample, which can be determined with a given probability [43,44]. When the concentrations increased from 10,000 to 90,000 ppm, the values of the detection limit decreased from 1213.33 to 2542.5. These values are relatively acceptable, which characterizes the proposed sensor in this study from previously designed liquid sensors of a comparable type (1D) and dimensions [43,44].

Accordingly, a brief comparison between the proposed sensor’s performance with previous PnC liquid sensors of a similar type and dimension is summarized in Table 2. As demonstrated in Table 2, the suggested sensor has a much higher sensitivity (1905 Hz/ppm) than several previously reported PnCs liquid sensors in part per million concentrations. Furthermore, the suggested cadmium bromide sensor in this study can be very selective while also being sensitive to the cadmium bromide solution. Other performance parameters are also quite acceptable and unique, as shown in previous figures. As a result, the proposed sensor functions more efficiently with particularly small concentrations, which are in ppm. Moreover, the proposed designed sensor can be better than other types of sensors, such as electrochemical sensors, which have a short or limited shelf life. The designed phononic sensor has several advantages. It is produced from readily available and less expensive materials, and it is simple to construct without the need for any electronic components.

### 3.6. Experimental Framework

This subsection presents the ability of an experimental realization of the proposed sensor. The proposed 1D-PnC design is essentially introduced based on a lead and epoxy multilayer periodic structure. The reasons behind choosing these materials are due to the distinctive utilization of these materials in PnC structures on both theoretical and experimental levels [5,6,10]. In addition, Gorishnyy et al. manufactured a PnC design based on the interference lithography technique [48]. This technique could be of potential interest because it provides easy control over the geometrical parameters such as the lattice length scale and volume fraction [49]. In addition, Villa-Arango et al. utilized transient time response for the characterization of a similar 1D PnC sensor [50]. In this context, a solution of lithium carbonate was used as a defect layer between 4-unit cells [51]. Then, two ultrasonic transducers made of a piezoelectric material were demonstrated to generate and receive the mechanical waves [52]. The authors used a Tektronix oscilloscope, an Analog Devices AD8302 integrated circuit, and a Tektronix precision signal generator to generate the signal and obtain the activity in response to the system for signal generation and collection. A USB cable was used to link all the apparatus to the computer using an Atmel ATmega2560 microcontroller. To integrate the input and output and obtain a rapid response of the system, a Python code was created to regulate both the generation and the collection of the signal [7,53]. After that, the transmitter was attached to the Tektronix generator and the receiver to the AD8302 integrated circuit together with a constant gain as a reference generator. The Python code was run to check if the signal is accurate. The sample was carefully put in the crystal’s central layer and was ready for use once the sensor and instruments were completely operational. The phononic crystal was meticulously cleaned and dried before each measurement, which was carried out three times in total. Since the temperature variation can influence the acoustic properties of the materials, which can, in turn, affect the results of the study [50,53], testing was completed in a temperature-controlled environment. To remove extra noise during signal processing, the signals were captured and filtered using a second-order, low-pass Butterworth filter with a cutoff frequency of 400 Hz. To prevent distorting the morphology of the generated frequency spectra, a high value was chosen as the filter’s cutoff frequency [50,54]. All the above steps and experimental measurements can be exactly applied to the proposed design, and the acoustic signal and transmission coefficient can be measured and detected easily.

## 4. Conclusions

In this paper, a 1D-PnC design with a defect layer filled with cadmium bromide is introduced. Depending on the transfer matrix method, the transmission spectrum was calculated for the acoustic waves’ propagation by using the proposed PnC design. The appearance of a resonant peak through the phononic bandgap represents the mainstay towards the detection of cadmium bromide ions in fresh water. The numerical findings demonstrated the optimum thicknesses of the cadmium bromide, lead, and epoxy layers, along with the effect of the number of periods. Based on the optimum condition, the experimental findings obtained a relatively high sensitivity of 1904.25 Hz, a Q-factor of 1528.76, and a figure of merit of 4.12088 × 10^−5^ (/ppm) as the concentration of cadmium bromide changed between 0 and 90,000 ppm. Finally, the output results could provide an easy technique for measuring low concentrations of cadmium bromide ions in fresh water and other heavy metals using cost-effective and versatile materials.

## Figures and Tables

**Figure 1 micromachines-14-00204-f001:**
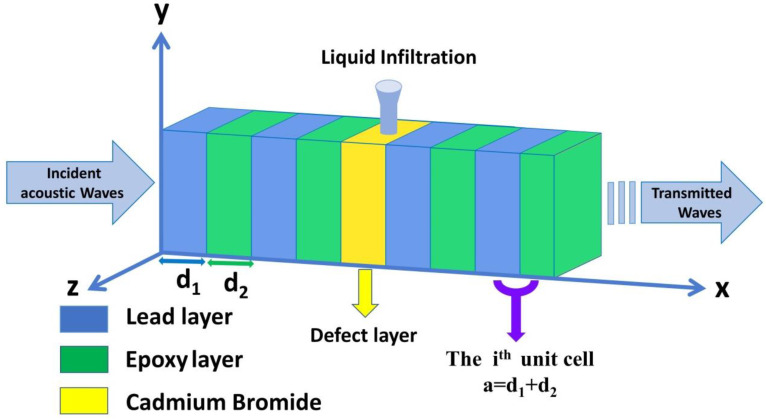
A schematic diagram of a defective 1D PnC structure designed from lead and epoxy with a defect layer filled with fresh water contaminated with cadmium bromide ions.

**Figure 2 micromachines-14-00204-f002:**
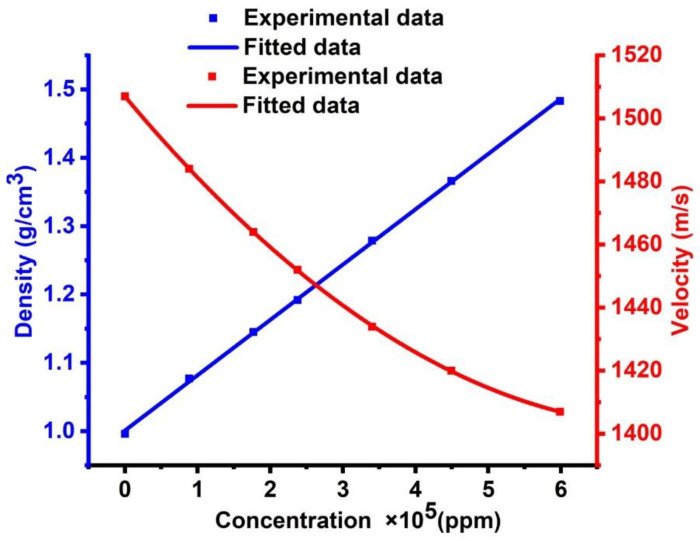
The effect of cadmium bromide concentration on its acoustic properties.

**Figure 3 micromachines-14-00204-f003:**
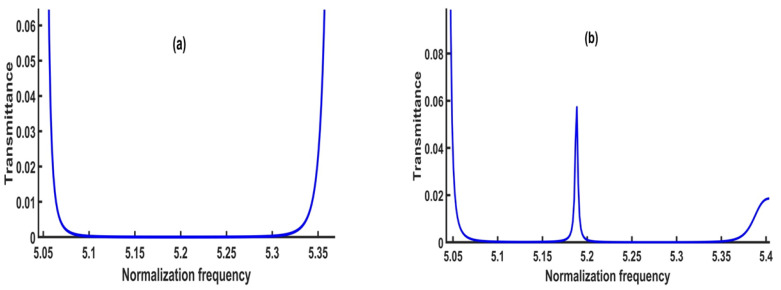
The transmission spectrum of (**a**) the perfect 1D PnC structure and (**b**) the 1D defective PnC structure.

**Figure 4 micromachines-14-00204-f004:**
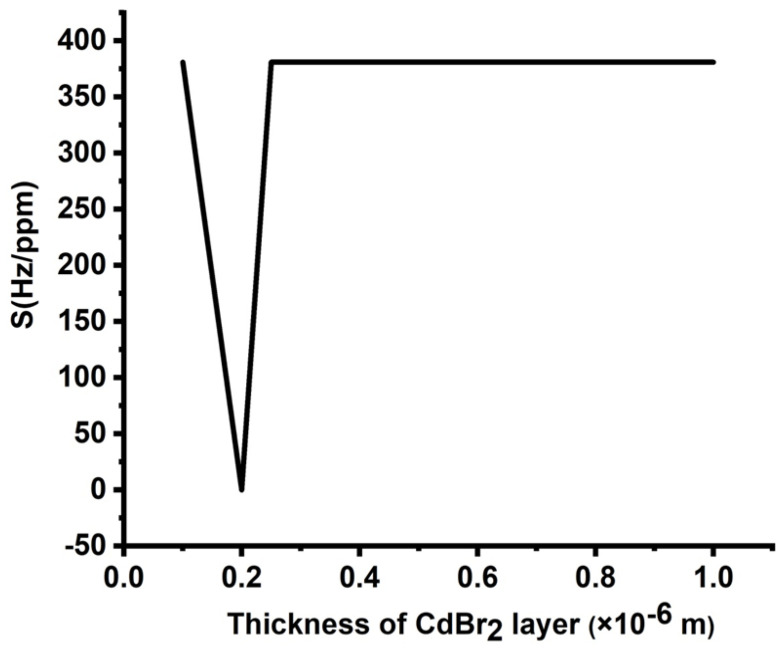
Sensitivity of the designed sensor at different thicknesses of the CdBr_2_ layer.

**Figure 5 micromachines-14-00204-f005:**
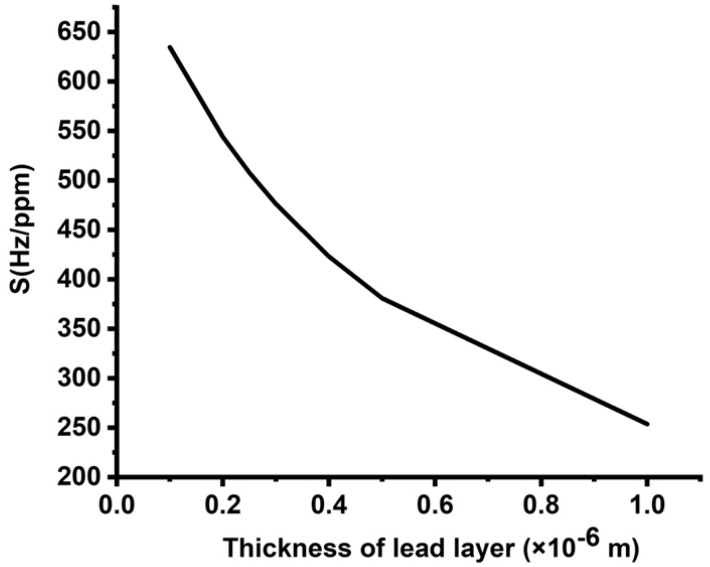
Sensitivity of the designed sensor at different thicknesses of lead layers.

**Figure 6 micromachines-14-00204-f006:**
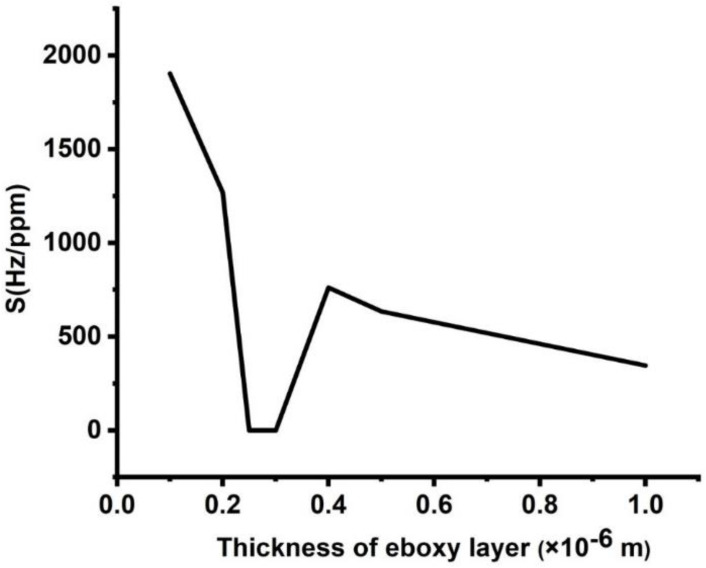
Sensitivity of the designed sensor at different thicknesses of the epoxy layer.

**Figure 7 micromachines-14-00204-f007:**
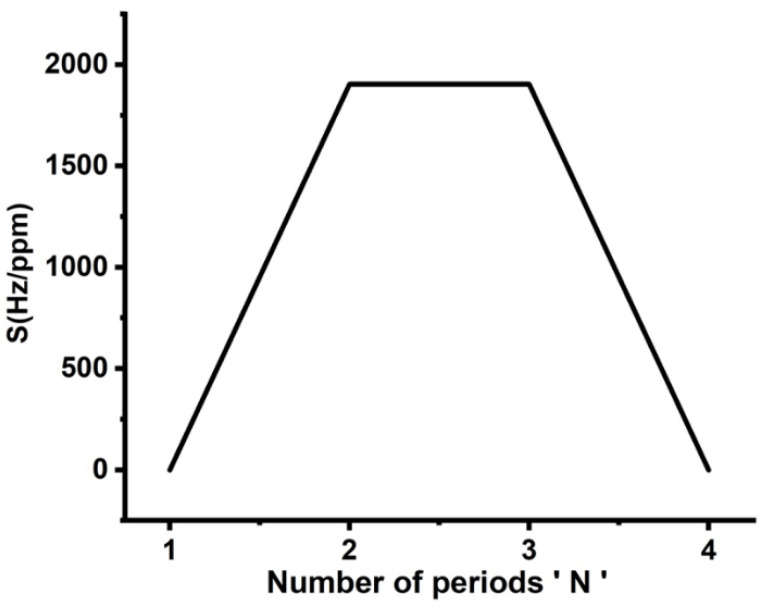
Dependence of the proposed sensor’s sensitivity on the number of periods of 1D-PnC.

**Figure 8 micromachines-14-00204-f008:**
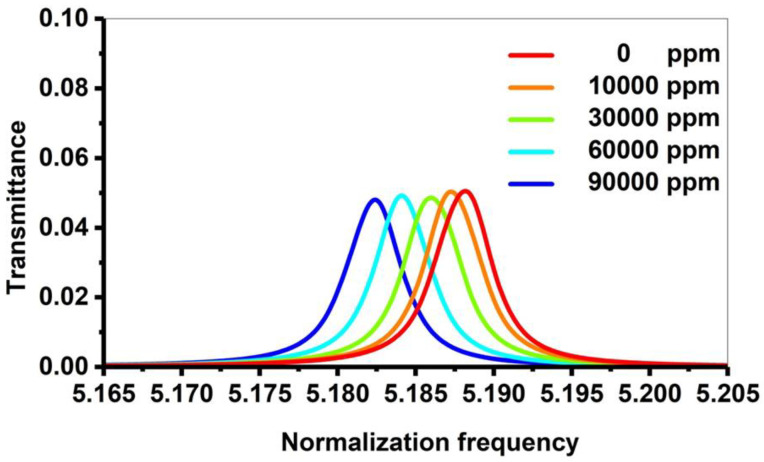
The transmission spectrum of the 1D PnC structure at different concentrations of cadmium bromide.

**Figure 9 micromachines-14-00204-f009:**
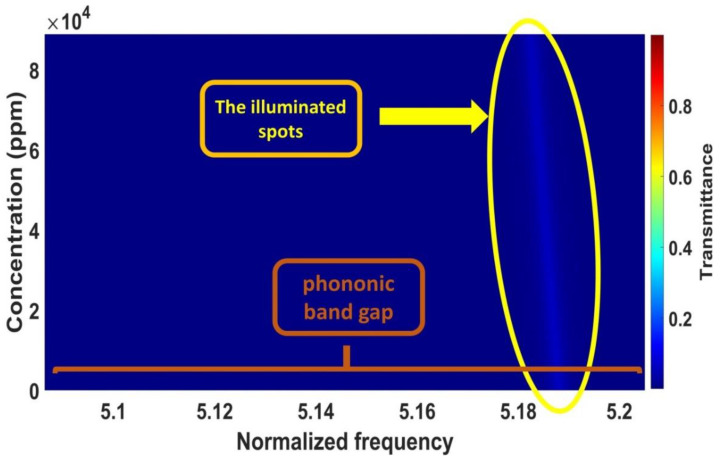
The color map of the transmission distribution versus cadmium bromide concentrations.

**Figure 10 micromachines-14-00204-f010:**
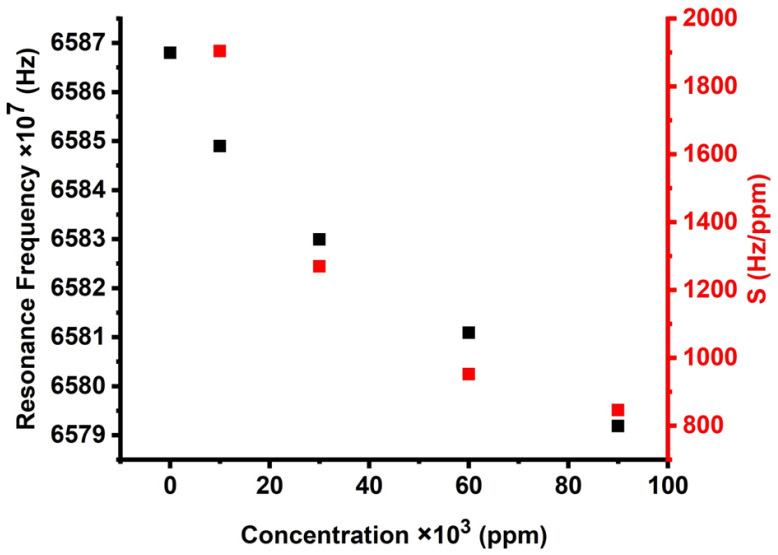
The effect of different concentrations on the resonance peaks and sensitivity of cadmium bromide liquid sensor.

**Figure 11 micromachines-14-00204-f011:**
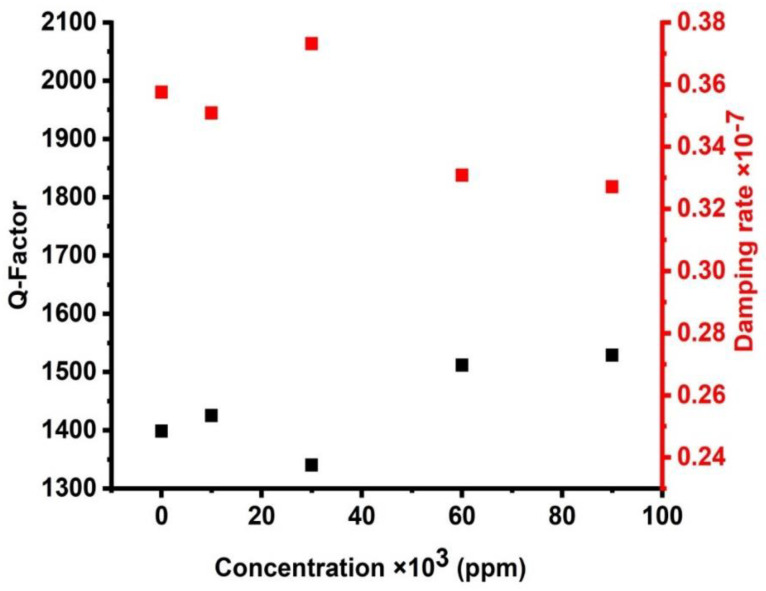
The effect of cadmium bromide concentrations on the quality factor and damping rate of the cadmium bromide sensor.

**Figure 12 micromachines-14-00204-f012:**
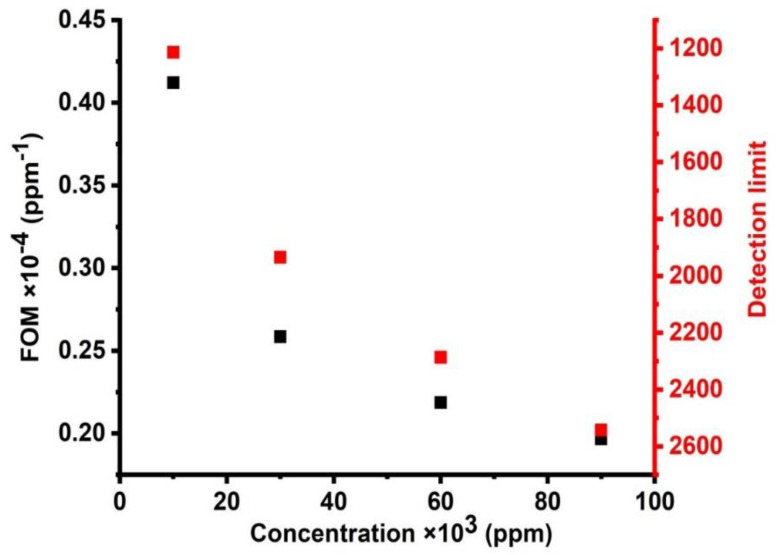
The effect of different concentrations on the figure of merit and detection limit of cadmium bromide liquid sensor.

**Table 1 micromachines-14-00204-t001:** Densities and speeds of the suggested materials’ sound.

Material	Mass Density ρ × 10^3^ (kg/m^3^)	Sound Speed c (m/s)
Lead	11.400	1960.12
Epoxy	1.18	2539.518
Water	0.998	1483

**Table 2 micromachines-14-00204-t002:** Comparison of the sensitivity of the proposed design with previously published works.

Type of Sensor	Sensing Material	Sensitivity	Reference
1D phononic crystal from Aluminum and epoxy multilayer with defect layer from biodiesel fuel	Methyl soy ester Oxidized soy ester Ethyl soy ester Certified D-2 Methyl laurate	34.14 Hz/m s^−1^ 35.39 Hz/m s^−1^ 36.02 Hz/m s^−1^ 43.88 Hz/m s^−1^ 50.37 Hz/m s^−1^	[40]
Locally resonant porous phononic crystal sensor for heavy metals detection: A new approach to highly sensitive liquid sensors	CdBr_2_	47.25 Hz/ppm	[45]
Glycine sensor based on 1D defective phononic crystalStructure	Glycine liquid at molar ratio range from 0 to 2 mol/L	969,973 Hz	[41]
3D-printed phononic fluidic cavity sensor	NaCl	(Experimentally) 1.4 kHz	[46]
The theoretical design of phononic crystal cavity sensor for simple and efficient detection of low concentrations of heavy metals in water	Cu(NO_3_)_2_	20 Hz/ppm	[47]
1D phononic crystal made of lead and epoxy multilayers with a cadmium bromide defect layer	CdBr_2_	1904.25 Hz/ppm	[The present work]

## Data Availability

Requests should be addressed to corresponding author.

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
