# Peer review of "One-Dimensional Phononic Crystals: A Simplified Platform for Effective Detection of Heavy Metals in Water with High Sensitivity"

_micromachines, 2023, doi:10.3390/mi14010204_

Round 1

Reviewer 1 Report

The authors must describe in more detail how they did the experiments:

For example, in figure 2 - how many measurements were made for each value. Did the temperature influence the acoustic properties? What is the value of repeatability?

Some of the simulations data presented in this paper should also be supported by experimental data.

Author Response

Authors' Response to Reviewers’ comments

Manuscript ID: Micromachines-2120214

Title: One-dimensional phononic crystals: A simplified platform for effective detection of heavy metals in water with high sensitivity

Journal: Micromachines

Abdulkarem H. M. Almawgani, Hamza Makhlouf Fathy, Hussein A. Elsayed, Ghassan Ahmed Ali and Ahmed Mehaney

Dear Editorial Board of Micromachines Journal:

Thanks a lot for your efforts in reviewing our manuscript. We’d like also to thank the reviewers for either their encouraging words of potential acceptance or their valuable comments and recommendations.

We have made our best to modify our article to match the reviewers' comments. In accordance, further explanation and comments have been added. In addition, the results and discussion section has been improved with some new physical aspects. Moreover, the English language of this paper was revised carefully to avoid any grammatical errors, unsuitable linkers and typos as well.

All changes have been underlined and re-written in red within the body of our revised manuscript.

In the following, we address a specific point by point response to the reviewers' comments:

Response to Reviewer 1:

The authors must describe in more detail how they did the experiments:

For example, in figure 2 - how many measurements were made for each value. Did the temperature influence the acoustic properties? What is the value of repeatability?

Some of the simulations data presented in this paper should also be supported by experimental data.

 Authors' response:

We would like to present our deep thanks for your efforts in reviewing our work. Actually, our work demonstrated a theoretical study for the detection of heavy metals concentrations through freshwater. The mainstay of this study is essentially depending on the well-known one-dimensional phononic crystals due to its simplicity, high efficiency, and low cost of fabrication as well. In this regard, the concentration of Cadmium Bromide in freshwater is considered for the sensing and detection based our suggested sensing tools. Here, the change in the concentration of CdBr2 in freshwater could lead to a significant change on the mechanical characteristics of freshwater (density and velocity) as investigated in Figure (2). This figure was fitted based on some previous experimental work as cited in the paper and you can see this unique reference of the experimental results of sound velocity in liquids to ensure from the correctness of data in Fig.2  [Schaafs, W.: Molekularakustik. Springer-Verlag, Berlin (1963)]. For your comment about temperature, we investigated all of our numerical findings at room temperature (the temperature of water in rivers and freshwater). Also, the water sample was not proposed to be heated above the room temperature. In addition, the lead layers at both sides prevent any increment of the design temperature from the surrounding atmosphere. Furthermore, the density of water decreases from 999.7 kg/m3 at 10 0C to only 992.25 kg/m3 at 40 0C [Mechanical Systems and Signal Processing 185 (2023) 109763]. This small value of density decrement at this wide range of temperature change, and indeed the room temperature lies definitely within the average of this range, will not affect the performance of the sensor. In particular, the considered sensor is introduced to operate at room temperature to be compatible with the real environment. Notably, the increase of temperature could lead to very small changes in the mechanical properties of the constituent materials especially at higher temperatures [https://www.worldscientific.com/doi/abs/10.1142/S0217979213500471]. Therefore, we believe that the role of temperature on the performance of our suggested sensor is very limited, especially at room temperature. For your recommendation about the support of our simulation results with some experimental data, figure (2) is essentially prepared based on some previous experimental data as mentioned above. In addition, we added a new subsection to discuss the experimental framework of the considered design.  

Reviewer 2 Report

The article by Fathy et al. titled:

'One-dimensional phononic crystals: A simple platform for effective detection of heavy metals in water with high sensitivity'

presents thereotical calculations and comparison with experimental data that show a new design of contamination detectors based on phononic crystals. I consider the work is interesting and suitable the journal Micromachines.

However, before suggesting acceptance it will be useful to clarify 4 issues:

1) Figure 9 does not look informative. Can the Authors tune the scale, color and discuss more its content? 

2) In Table 2 the sensitivity is given in different units for different works. Can the Authors comment on that, discuss how to compare different values and -if possible- give the same units for all?

3) In Figure 4 I have the question why the sensitivity is dropping so strongly for a specific thickness (0.2 micrometers) and other than that it is constant.

4) Can the Authors provide additional discussion on the potential experimental realization of their device? (method, feasibility, role of fabrication imperfections).

Author Response

Authors' Response to Reviewers’ comments

Manuscript ID: Micromachines-2120214

Title: One-dimensional phononic crystals: A simplified platform for effective detection of heavy metals in water with high sensitivity

Journal: Micromachines

Abdulkarem H. M. Almawgani, Hamza Makhlouf Fathy, Hussein A. Elsayed, Ghassan Ahmed Ali and Ahmed Mehaney

Dear Editorial Board of Micromachines Journal:

Thanks a lot for your efforts in reviewing our manuscript. We’d like also to thank the reviewers for either their encouraging words of potential acceptance or their valuable comments and recommendations.

We have made our best to modify our article to match the reviewers' comments. In accordance, further explanation and comments have been added. In addition, the results and discussion section has been improved with some new physical aspects. Moreover, the English language of this paper was revised carefully to avoid any grammatical errors, unsuitable linkers and typos as well.

All changes have been underlined and re-written in red within the body of our revised manuscript.

In the following, we address a specific point by point response to the reviewers' comments:

Response to Reviewer 2:

The article by Fathy et al. titled:

'One-dimensional phononic crystals: A simple platform for effective detection of heavy metals in water with high sensitivity' presents thereotical calculations and comparison with experimental data that show a new design of contamination detectors based on phononic crystals. I consider the work is interesting and suitable the journal Micromachines.

However, before suggesting acceptance it will be useful to clarify 4 issues:

1) Figure 9 does not look informative. Can the Authors tune the scale, color and discuss more its content?

Authors' response:

Thank you for your useful comments concerning our manuscript and your suggestions for this work. For this comment, we modified this figure again with more clarifications and discussions as well, see the below discussions in red in page 19.

Meanwhile, figure (9) shows a color map that simulates the Appearance of the resonant peak within the formed phononic band gap due to the increase in the concentration of cadmium bromide and the movement of the resonant peak through the phononic band gap. To sum up, the transmission distribution of the acoustic waves in the defected PnC versus the normalized frequency for a concentration range of 0–90 × 103 ppm is shown in Fig. 9. Here, the illuminated spots (Highest transmission values; resonant modes) in the normalized frequency range (5.18–5.2) shifted with increasing different concentrations; this means that the illuminated line has a very clear slope. This means the normalized frequency values (indicated in the x-axis) changing against different concentrations (indicated in the left y-axis), which, in turn, confirms the obtained results in figure (8). In addition, different shades of colors indicate different percentages of the intensity of the transmitted waves and the high sensitivity of the proposed sensor versus the concentration in the ppm scale.

2) In Table 2 the sensitivity is given in different units for different works. Can the Authors comment on that, discuss how to compare different values and -if possible- give the same units for all?

Authors' response:

Thanks a lot for your observation; we tried to find the most comparable researches with the same sensitivity units, but we could not convert some units as the authors depend in calculating the sensitivity on a different input parameter like the different sound speeds of some liquids (Here we used the concentration of heavy metals in ppm scale). We mentioned these works as they are all liquid phononic sensors and to just highlight the differences between our design and them although the input parameters is different.  However, we modified table 2 with new works that used the same sensitivity unit.    

3) In Figure 4 I have the question why the sensitivity is dropping so strongly for a specific thickness (0.2 micrometers) and other than that it is constant.

Authors' response:

Thank you for this note, the blew explanation for this comment is added I n the paper, page 14.

Actually, the unusual decrease in the sensitivity value at a thickness of 0.2 μm may be attributed to the condition of constant phase shift at this thickness value. This explanation is physically based on Bragg’s diffraction law which is considered the main core of the band gap and resonant modes formation [3,36]. At this value definitely, the sensitivity drops to approximately the zero value, this means the resonance peak position is nearly unchanged at this thickness value. Based on the above-mentioned Bragg’s diffraction law, at this thickness value, the same type and order of interference occurred at this value, so the peak position did not change. 

4) Can the Authors provide additional discussion on the potential experimental realization of their device? (method, feasibility, role of fabrication imperfections).

Authors' response:

Thanks a lot for your useful comment. Meanwhile, we added a new subsection through the results and discussions section to highlight the possibility of the experimental realization.

We hope this response meets the reviewers' comments and their points of view. In case the reviewers have any other comments, we are willing to carry out them all to meet their requirements. Looking forward to hearing of acceptance of our contribution.

With respect in advance;

The corresponding author;

Round 2

Reviewer 1 Report

Although the authors' answers were not to the point, the paper can be accepted in its current form.